# Anesthetics and Cell–Cell Communication: Potential Ca^2+^-Calmodulin Role in Gap Junction Channel Gating by Heptanol, Halothane and Isoflurane

**DOI:** 10.3390/ijms23169017

**Published:** 2022-08-12

**Authors:** Camillo Peracchia

**Affiliations:** Department of Pharmacology and Physiology, School of Medicine and Dentistry, University Rochester, 601 Elmwood Avenue, Rochester, NY 14642, USA; camillo.peracchia@gmail.com

**Keywords:** anesthetics, heptanol, halothane, isoflurane, gap junctions, connexins, channel gating, calcium, calmodulin, cell communication, cell-to-cell channels, cell coupling, cell uncoupling, membrane channels

## Abstract

Cell–cell communication via gap junction channels is known to be inhibited by the anesthetics heptanol, halothane and isoflurane; however, despite numerous studies, the mechanism of gap junction channel gating by anesthetics is still poorly understood. In the early nineties, we reported that gating by anesthetics is strongly potentiated by caffeine and theophylline and inhibited by 4-Aminopyridine. Neither Ca^2+^ channel blockers nor 3-isobutyl-1-methylxanthine (IBMX), forskolin, CPT-cAMP, 8Br-cGMP, adenosine, phorbol ester or H7 had significant effects on gating by anesthetics. In our publication, we concluded that neither cytosolic Ca^2+^_i_ nor pH_i_ were involved, and suggested a direct effect of anesthetics on gap junction channel proteins. However, while a direct effect cannot be excluded, based on the potentiating effect of caffeine and theophylline added to anesthetics and data published over the past three decades, we are now reconsidering our earlier interpretation and propose an alternative hypothesis that uncoupling by heptanol, halothane and isoflurane may actually result from a rise in cytosolic Ca^2+^ concentration ([Ca^2+^]_i_) and consequential activation of calmodulin linked to gap junction proteins.

## 1. Introduction

Direct cell–cell communication is mediated by gap junction channels that enable free exchange of small cytosolic molecules among neighboring cells. Each channel is formed by two hemichannels that create a hydrophilic passageway, spanning the plasma membranes of two neighboring cells and a narrow extracellular space (gap). A gap junction hemichannel is made of six radially arranged proteins named connexins in vertebrates and innexins in invertebrates. Connexins/innexins contain two extracellular lops (EL_1_ and El_2_) and three cytoplasmic domains: an NH_2_-terminus (NT), a cytoplasmic loop (CL) and a COOH-terminus domain (CT; rev, in [1]).

Gap junction channels are physiologically regulated by a chemical gating mechanism that is activated by changes in cytosolic ionic homeostasis, resulting from cell damage, inhibition of the metabolism, acidification and hypoxia, among others. Over the years, numerous studies have suggested that channel gating results from a rise in cytosolic calcium concentration ([Ca^2+^]_i_) [2,3,4] in the high nanomolar–low micromolar range [1,5,6,7]. Since the early 1980s, we have proposed that Ca^2+^_i_ causes gating by activating calmodulin (CaM) [1,7,8,9,10] via a cork-like pore-plugging mechanism [11,12,13] probably involving conformational changes in connexins as well.

In the last four decades, many studies have reported that gap junction channels are sensitive to anesthetics. Channel gating induced by long-chain alcohols (heptanol and octanol) was first reported by Johnston and coworkers [14] in crayfish septate axons. Soon after, gating by volatile anesthetics such as halothane and isoflurane was also reported in a variety of vertebrate and invertebrate systems [15,16].

In spite of over four decades of research, the mechanism by which anesthetics cause gap junction channel gating is still poorly understood. Johnston and coworkers [14] proposed an extracellular site of action for alkanols. Similarly, Eskandari and coworkers [17] reported that, in inside-out patches of lens connexin hemichannels, the addition of 1 mM octanol did not affect the channel’s open probability or the unitary conductance, while in outside-out patches, addition of 1 mM octanol to the bath (extracellular surface of hemichannels) significantly reduced single-channel open probability without altering the unitary current. Therefore, they concluded that octanol inhibits lens connexin hemichannels by acting on a site accessible only from the extracellular space. However, the possibility that a soluble intermediate was washed away by the internal perfusion of crayfish axons [14] and by the bathing solution of inside-out patches [17] was not considered.

In 1991, our data seemed to also exclude the role of cytosolic Ca^2+^ and pH in crayfish axons uncoupled by heptanol, halothane and isoflurane [16]. However, based on the potentiating effect of caffeine and theophylline added to anesthetics, and the numerous studies published in the past three decades on the effect of anesthetics on both CaM’s Ca^2+^ sensitivity and Ca^2+^ release from the sarcoplasmic reticulum (SR) via inositol trisphosphate (IP3) and/or ryanodine receptor (RyR) channels, as well as evidence that ion-selective electrodes may be unreliable in the presence of anesthetics [18], we are now reconsidering our earlier interpretation [16]. As a result, we propose an alternative hypothesis, that cell–cell uncoupling by anesthetics may result from CaM’s activation caused by anesthetic-induced [Ca^2+^]_i_ rise, and increased Ca^2+^ sensitivity in CaM.

## 2. Measurement of Junctional Resistance (Rj), [Ca^2+^]_i_ and [H^+^]_i_ in Crayfish Septate Axons

In our 1990s study [16], crayfish axons were superfused with a standard saline solution for crayfish (SES) [19], containing (in mM): NaCl, 205; KCl, 5.4; CaCl_2_, 13.5 and 4-(2-hydroxyethyl)-1-piperazineethanesulfonic acid (HEPES), 5 (pH 7.5). Either 2.8–5.6 mM 1-heptanol, 9.5–28.5 mM halothane or 23.6 mM isoflurane was added to SES in the presence and absence of either 10–20 mM caffeine or 10-20 mM theophylline. For testing the effect of low pH_i_ [20,21], the axons were superfused with a sodium acetate saline solution (Ac) containing (in mM): Na acetate, 205, KCI, 5.4 and CaC1_2_, 13.5 (pH 6.3).

Four microelectrodes were inserted into a lateral giant axon, two on each side of the septum (Figure 1A), and hyperpolarizing square current pulses (150 nA, 300 ms) were passed every 10 s alternatively into the posterior (C_1_) and anterior (C_2_) axon segments. The resulting electrotonic potentials V_1_ and V_2_ (from current injection in C_1_), V_1_* and V_2_* (from injection in C_2_) and the membrane potentials (E_1_ and E_2_) were recorded with two voltage microelectrodes through a voltage follower. The voltage signals were displayed on an oscilloscope and a chart recorder, and were digitized. Both membrane (R_m1_, R_m2_) and junctional (Rj_1_, Rj_2_) resistances were calculated from current (I_1_, I_2_) and voltage (V_1_,V_2_,V_1_*,V_2_*) records (Figure 1B) [16].

[Ca^2+^]_i_ and [H^+^]_i_ were measured with ion-sensitive microelectrodes based on neutral-carrier sensors. Ca^2+^ microelectrodes used the calcium cocktail (ETH 129) and H^+^ microelectrodes used the proton cocktail tri-n-dodecylamine. At the time, we thought that these ion-selective microelectrodes were insensitive to anesthetics. Later on, however, ion-selective microelectrodes were proven unreliable in the presence of anesthetics [18]. In contrast, these ion-selective microelectrodes are reliable for measuring changes in [Ca^2+^]_i_ or [H^+^]_i_ caused by acetate-induced cytosolic acidification [20,21].

## 3. Gating by Heptanol in the Presence and Absence of Caffeine or Theophylline

Lateral giant axons have a membrane potential that ranges from −80 to −95 mV (Figure 2A and Figure 3A) and are electrically coupled at the septum with an initial Rj of 150 ± 53.7 kΩ (mean ± SD; n = 28). Superfusion of the axons with 2.8–5.6 mM heptanol results in a small depolarization (Figure 2A and Figure 3A) and increases Rj by 191.3 ± 83% (mean ± SD; n = 28) of control values (Figure 2B and Figure 3B). Note that the peaks of depolarization and Rj correspond well (Figure 2A,B). Recovery of both Rj and membrane potential (MP) are quicker than their onset (Figure 2A,B and Figure 3B).

Addition of caffeine (10–20 mM) to heptanol solutions increases Rj maxima by 309.3 ± 265% (mean ± SD; n = 24) of controls treated with heptanol alone (Figure 2B and Figure 3B) and causes greater depolarization (Figure 2A and Figure 3A). Significantly, when caffeine is added to heptanol several minutes after the beginning of heptanol treatment, the rates of depolarization (Figure 3A) and Rj rise (Figure 3B) greatly increase. Indeed, a 7 min superfusion of heptanol–caffeine, following a 22 min superfusion of heptanol alone, virtually doubles the Rj rise induced by heptanol alone (Figure 3B). Following the first heptanol–caffeine application the Rj maxima with heptanol alone or heptanol–caffeine significantly decreased (Figure 4, blue and red arrows, respectively). This may suggest that the first heptanol–caffeine treatment reduced the calcium content of the endoplasmic reticulum (ER). Addition of 10 μM ryanodine partially reduces the Rj maxima reached with heptanol–caffeine (Figure 5), suggesting a partial inhibition of calcium release from the ER stores.

Addition of 10–20 mM theophylline to heptanol solutions also dramatically enhanced the heptanol effects on Rj (Figure 6). Indeed, Rj maxima with heptanol–theophylline are 676 ± 386% (mean ± SD; n = 4) greater than those with heptanol alone. Neither different external [Ca^2+^]_o_, ranging from 7 to 27 mM, nor blockers of Ca^2+^ entry, such as Cd^2+^ (500 μM) or nisoldipine (10 μM), significantly change the heptanol-uncoupling efficiency [16].

The possibility that the effect of caffeine or theophylline is due to an increase in cyclic nucleotides was tested by exposing the axons to 3-isobutyl-1-methylxanthine (IBMX, a phosphodiesterase inhibitor), forskolin (an activator of adenylate cyclase) or diffusible cAMP and cGMP (CPT-cAMP and 8Br-cGMP) [16]. Additions to heptanol of 1 mM IBMX (Figure 7), a phosphodiesterase inhibitor 200 times more potent than caffeine [22], 5 μM forskolin, 500 μM CPT-cAMP or 200 μM 8Br-cGMP do not significantly affect Rj maxima [16]. The possible involvement of protein kinase C (PKC) was tested by superfusing the axons with heptanol solutions containing either 162 nM TPA (4β-phorbol-12β-myristate-13α-acetate) or 100 μM H7 (1-(5-isoquinoliny sulfonyl)-2-methylpiperazine); neither TPA (an activator of PKC) nor H7 (an inhibitor of PKC) significantly affected the magnitude of heptanol-induced uncoupling [16].

In view of the fact that caffeine is also a powerful inhibitor of adenosine receptors, the effect of adenosine on Rj was tested both in the presence and absence of heptanol. Superfusion of 1.3–5 mM adenosine, added to either SES or heptanol solutions, does not significantly change either control Rj values or Rj maxima with heptanol, indicating that adenosine receptors do not participate in the effect of heptanol and caffeine on gap junction channel conductance [16]. Lack of an involvement of adenosine receptor inhibition is also provided by the absence of an effect of IBMX on heptanol-induced Rj rise (Figure 7) [16]; indeed, IBMX, as caffeine and theophylline, is also an inhibitor of adenosine receptors [23,24].

Curiously, the K^+^ channel blocker 4-aminopyridine (4-AP) strongly inhibits the heptanol-induced uncoupling. With heptanol solutions containing 5 mM 4-AP, the Rj maxima are 26.2 ± 20% (mean ± SD; n = 12) lower than those with heptanol alone (Figure 8A). In contrast, addition of 4-AP (5 mM) to acetate solutions does not alter their uncoupling effects (Figure 8B). No effect on Rj was seen with 4-AP alone, with the only change being a 3–4 mV depolarization [16], probably caused by the inhibition of K^+^ channels.

## 4. Gating by Halothane or Isoflurane in the Presence and Absence of Caffeine

Superfusion of crayfish axons with halothane (28.5 mM) causes a small depolarization (Figure 9A) and increases Rj by 155.6 ± 56% (mean ± SD; n = 9) of control values of 226 ± 73 kΩ (mean ±SD; n = 9) (Figure 9B and Figure 10) [16]. Addition of 20 mM caffeine to halothane solutions causes a more marked depolarization (Figure 9A) and a greater increase in Rj (Figure 9B and Figure 10). The Rj maxima with halothane–caffeine are 329 ± 147% (mean ± SD; n = 8) greater than those with halothane alone [16].

Similar results were obtained with isoflurane (23.6 mM), which causes a 4–5 mV depolarization (data not shown) and increases Rj by ~125% of control values (Figure 10) [16]. With addition of 20 mM caffeine to isoflurane solutions, Rj maxima are ~170% greater than those with isoflurane alone (Figure 10).

## 5. Potential Mechanism of Channel Gating by Anesthetics

In our 1991 study, we concluded that the uncoupling mechanism of the anesthetics heptanol, halothane and isoflurane involves a direct interaction between anesthetics and amphiphilic chains of gap junction proteins. Our interpretation was based primarily on the apparent drop in [Ca^2+^]_i_ and [H^+^]_i_. However, as mentioned in the previous section, after our publication, ion-selective electrodes have been reported not to be very reliable in the presence of anesthetics [18], such that a role of Ca^2+^_i_ could not be excluded. Indeed, we are now reconsidering our earlier interpretation and propose a hypothesis suggesting the potential role of Ca^2+^-activated CaM.

### 5.1. Does Uncoupling by Anesthetics Result from Increased [Ca^2+^]_i_ Potentiated by Caffeine and Theophylline?

Meda and coworkers [25] reported no significant change in [Ca^2+^]_i_ measured with Quin-2, in exocrine pancreas treated with heptanol or octanol at concentrations that greatly increased amylase release. However, this is curious because amylase release is known to be caused by a sustained [Ca^2+^]_i_ rise [26,27,28]; therefore, one may wonder whether the Ca^2+^ sensitivity of Quin-2 might have been affected by alkanols. In agreement with this report [25], some studies have reported the normal uncoupling efficiency of alkanols and halothane with patch pipette solutions buffered for Ca^2+^ [29,30,31,32]. However, these data conflict with several other studies which reported an alkanol-induced [Ca^2+^]_i_ rise. In a careful double whole-cell patch-clamp study on embryonic chick cardiac cells, Veenstra and DeHaan [33] reported that octanol uncoupled the cells by reducing junctional conductance (Gj) to 4.3 ± 3.4% of the control level with patch pipette solutions weakly buffered for Ca^2+^ (0.1 mM EGTA), but octanol reduced Gj by only 29.0 ± 19.1% with more strongly Ca^2+^-buffered solutions (5 mM EGTA), in their words [33]: “*A possible mechanism for the octanol effect comes from studies by Vassort* et al. ([34]) *indicating that long-chain alcohols inhibit cytoplasmic Ca-buffering. These workers reported that octanol caused a rise in [Cai]c* (cytosolic [Ca^2+^]) *in squid axons, associated with an alkalinization of the axoplasm, and they presented evidence that Ca^2+^ is released from an intracellular binding site in exchange for H^+^ ions …. these observations are consistent with our finding that octanol uncoupled cell pairs when calcium buffering was minimal, and that the effect was reduced when the cells were infused with Lo-Ca intracellular pipette solutions containing 5 mM EGTA*”.

Furthermore, halothane and n-alkanols increased resting [Ca^2+^]_i_ by 20–70% in mouse whole-brain synaptosomes [35], and several studies reported a [Ca^2+^]_i_ rise, monitored by the Ca^2+^-indicator arsenazo III, in octanol-treated squid axons [34,36,37]. In addition, a study on smooth muscle cells of porcine airway reported that halothane increases [Ca^2+^]_i_ by a Ca^2+^ leak through both inositol (1,4,5)-trisphosphate-(IP3)- and ryanodine-receptor channels (RyR) [38]. In pancreatic acinar cells, halothane and octanol induced a sustained [Ca^2+^]_i_ increase [39]. In a study on malignant hyperthermia (MH), halothane, used in contracture testing for MH susceptibility, caused large elevations of myoplasmic [Ca^2+^] [40]. In vascular smooth muscles, halothane causes both Ca^2+^ release from stores and stimulates Ca^2+^ uptake [41]; the halothane-induced Ca^2+^ release from the stores is sensitive to both caffeine and IP3, suggesting that both RyR and IP3 channels of the ER play a role [41]. In a more recent study, volatile anesthetics were found to cause cell damage by abnormal calcium release from the ER via excessive activation of IP3-receptor channels [42], and the anesthetics’ neuroprotective and neurotoxic mechanisms have been shown to involve Ca^2+^ release from the ER’s IP3-receptor channels [43]. In frog skeletal muscle, halothane was found to increase [Ca^2+^]_i_ by releasing it from the sarcoplasmic reticulum (SR) via the RyR’s Ca^2+^-release channel [44]. 

Morphological studies also reported changes in gap junction particle size and spacing with heptanol and other uncouplers known to increase [Ca^2+^]_i_ in heart [45,46] and pancreas [47]. While the structural changes in gap junctions most likely are due to Ca^2+^-activated CaM, it should be kept in mind that halothane has been reported to change the domain structure of a model membrane [48]. X-ray and neutron-diffraction studies of a binary lipid membrane demonstrate that halothane at physiological concentrations produces a pronounced redistribution of lipids between domains of different lipid types identified by different lamellar d-spacings and isotope composition. The redistribution of lipids between domains induced by anesthetics could in principle contribute to changes in gap junction particle size and spacing as well. Furthermore, gap junctions are rich in cholesterol [49,50,51,52]. Indeed, Bastiaanse and coworkers [53] have suggested that heptanol decreases gap junction channel conductance by decreasing the fluidity of cholesterol-rich domains in cardiac cells. Therefore, it is likely that anesthetics contribute to cell–cell uncoupling by affecting the fluidity of the gap junction bilayer as well. 

Plasma membranes of cells show asymmetric lipid distribution between the bilayer leaflets with a negative charge of the inner bilayer leaflet [54]. Phospholipid unsaturation is dramatically asymmetric, with the cytoplasmic leaflet being approximately twofold more unsaturated than the exoplasmic leaflet [55]. Atomistic simulations and spectroscopy of leaflet-selective fluorescent probes reveal that the outer PM leaflet is more packed and less diffusive than the inner leaflet. The tightly packed outer leaflet may serve as an effective permeability barrier, while the more fluid inner leaflet may allow for rapid signal transmission. Thus, it is conceivable that the solubility of anesthetics is different in the two halves of the membrane bilayer.

The effect of caffeine and theophylline on uncoupling by anesthetics also indicates a participation of Ca^2+^ release from stores, as previously shown with acidification [20,21]. Indeed, both caffeine and theophylline are known to increase [Ca^2+^]_i_ by releasing it from Ca^2+^ stores [56,57]. At mM concentrations, caffeine exerts a powerful effect on the SR by activating Ca^2+^ release via RyR channels and perhaps also by inhibiting calcium reuptake [57]. Remarkably, the uncoupling mechanism of halothane seems to parallel in some way the mechanism of MH (reviewed in [58]), as in both cases the halothane-induced Ca^2+^ release from the stores is potentiated by caffeine (see the caffeine–halothane contracture test). Perhaps, heptanol and halothane release Ca^2+^ from stores by activating either the IP3 or the RyR receptor (Figure 2A, Figure 3A, Figure 9A and Figure 11, inset b, green arrows), while the addition of caffeine or theophylline releases Ca^2+^ by activating both IP3 and RyR receptors (Figure 2A, Figure 3A and Figure 9A, green and red arrows).

Although, in crayfish axons, Ca^2+^ release from IP3- and/or RyR-receptor channels seems most likely, one should keep in mind that the effect of xanthines, such as caffeine and theophylline, and anesthetics on Ca^2+^ release from stores is complex and depends on cell type. Indeed, Bezprozvanny and coworkers [59] demonstrated caffeine-induced inhibition of IP3-gated calcium channels from cerebellum incorporated into planar lipid bilayers. Furthermore, Parker and Ivorra [60] found that caffeine inhibits IP3-mediated liberation of intracellular Ca^2+^ in *Xenopus* oocytes, and Saleem and coworkers [61] showed that caffeine is a low-affinity antagonist of type 1 IP3 receptors (IP3R1), while it had no significant effect on IP3-evoked Ca^2+^ release via IP3R2 or IP3R3.

Joseph and coworkers [62] found that isoflurane modulates IP3R channel sensitivity to IP3 only at low, sub-saturating concentrations of IP3 (<0.1 μM), and showed that isoflurane causes Ca^2+^ release from the ER via this activation of IP3R which can regulate intracellular Ca^2+^ homeostasis and apoptosis. In frog skeletal muscle, halothane was found to increase [Ca^2+^]_i_ by releasing it from the SR via the RyR’s Ca^2+^-release channel [44]. Later, Laver and coworkers [63] found that halothane activation of RyR2 is different from that seen in the skeletal isoform RyR1. Unlike RyR1, RyR2 was reported to be responsive to halothane and enflurane when recorded in bilayers [64] as well as in myocardial cells [65]. For gap junction channel gating in the heart, the cardiac RyR2 isoform is more relevant as halothane has been reported to activate the cardiac-ryanodine-receptor channel, while isoflurane proved ineffective in activating RyR2 [63,64].

The magnitude of the effect of anesthetics of gap junction channel gating is also related to the type of gap junction protein expressed. Indeed, He and Burt [66] have reported that halothane has only a small uncoupling effect in cells expressing human Cx40, but has a great effect in cell expressing heteromeric (human) Cx40/Cx43 channels. This is interesting and may support the Ca^2+^ role, because Xu and coworkers [67] reported that ionomycin application increases [Ca^2+^]_i_ and causes Gj to drop by 95% in N2a cells expressing human Cx43, but not in cells expressing human Cx40. Significantly, the human Cx40 does not have the CaM-binding site at the second half of the cytoplasmic loop (CL2), while Cx43 does [1,67].

The inhibitory action of the K^+^ channel blocker 4-aminopyridine (4-AP) on heptanol-induced gating (Figure 8A) is puzzling. It is interesting to note, however, that the potentiating effect of 4-AP on excitatory postsynaptic potentials (EPSP) in frog motoneurons is opposite to that of heptanol, which is an EPSP blocker [68,69]. Evidence that 4-AP fails to induce porcine MH [70], in spite of the fact that it mobilizes Ca^2+^ from intracellular stores [71,72], is also puzzling. Interestingly, 4-AP has been shown to modify phospholipid metabolism [73] and to stimulate protein phosphorylation in a Ca^2+^-dependent manner [74].

Indirect evidence of [Ca^2+^]_i_ rise with anesthetics is also provided by changes in membrane potential (MP). Indeed, the peak of membrane depolarization with heptanol (Figure 2A, Figure 3A and Figure 11, inset b) or halothane (Figure 9A), both alone (green arrows) or in the presence of caffeine (green and red arrows), matches the peak of Rj rise well (Figure 2B, Figure 3B and Figure 9B). Sauviat and coworkers [44] reported that the effects of halothane on membrane depolarization are likely to result from increased [Ca^2+^]_i_ due to Ca^2+^ release via the RyR channel, perhaps as well as by the activation of Ca^2+^-dependent Cl^-^ channels.

It is significant to compare the MP and Rj changes caused by anesthetics with those caused by cytosolic acidification, which is known to increase [Ca^2+^]_i_ [21,75] (Figure 11). While with anesthetics the MP change is monophasic (Figure 2A, Figure 3A, Figure 9A and Figure 11, inset b), with acidification it is biphasic (Figure 11A and inset a). The initial mild hyperpolarization with acidification is likely to result from Ca^2+^-activation of K^+^ channels [76]—it is absent with anesthetics probably because they inhibit K^+^ channels [77]. The depolarization with both anesthetics and acidification is most likely caused by Ca^2+^ release from stores.

### 5.2. Role of a Potential Soluble Intermediate—Calmodulin

Johnson and coworkers [14] proposed that alkanols act on an extracellular binding site, but did not consider the possible loss of a soluble intermediate, which might have been washed out by the internal perfusion of crayfish axons. Indeed, the standard internal solution (SIS) used contained (in mM): NaCl 15, K-fluoride 109, K-citrate 37, mannitol 96, HEPES 1, pH 7.5; the presence of K-fluoride and K-citrate might have lowered the [Ca^2+^]_i_ sufficiently to release CaM from binding sites and wash it away. Similarly, Eskandari and coworkers [17], who reported that octanol did not close lens connexin hemichannels in inside-out patches, did not consider the possibility that CaM was lost by exposing the cytosolic side of the membrane to solutions containing 5 mM EGTA and 1 mM CaCl_2_. Furthermore, the idea that molecules such as heptanol, octanol and halothane and other general anesthetics only bind to an extracellular membrane site is hard to conceive because these molecules easily cross the plasma membrane, such that their concentration on both sides of the membrane is rapidly equilibrated. Consistent with the idea that CaM may play a role in the gating by anesthetics both provides evidence that halothane binds to CaM [78] and provides evidence that halothane, isoflurane and alcohols increase CaM’s Ca^2+^ sensitivity [79,80].

On the other hand, Zhou and coworkers [81] showed that volatile anesthetics (VA) inhibit the activity of calmodulin by interacting with its hydrophobic core. Accordingly, Streiff and coworkers [82] predicted that volatile anesthetics bind to Ca^2+^-bound CaM (holo-CaM), but not to apo-CaM. The VA-binding sites predicted for the structures of holo-CaM are located in hydrophobic pockets that form when the Ca^2+^-binding sites in CaM are saturated. Volpi and coworkers [83] reported the antagonism of CaM by local anesthetics and the inhibition of the CaM-stimulated calcium transport of erythrocyte in an inside-out membrane vesicle. Levin and Blanck [79] observed a biphasic effect of halothane and isoflurane on calmodulin: at low concentrations of the anesthetics, the affinity of calmodulin for Ca^2+^ was decreased, while at higher concentrations, the affinity for Ca^2+^ was increased. Moreover, Rudnick et al. suggested that halothane mimics calmodulin-blocking agents and may alter CaM interaction with Ca^2+^-dependent kinases.

## 6. Summary and Conclusions

When we studied the effect of anesthetics on channel gating in crayfish lateral giant axons, we were puzzled by the apparent contradiction between the striking potentiating effect of caffeine and theophylline and the apparent drop in [Ca^2+^]_i_ [16]. Indeed, the effect of caffeine and theophylline on Rj and MP, and the fact that both the peaks of Rj rise and depolarization coincided, pointed to an increase in [Ca^2+^]_i_. The reason that now, three decades later, we are reevaluating the earlier interpretation of the lack of Ca^2+^_i_ participation in the effect of heptanol, halothane and isoflurane on gating is based on a number of facts, as follows:The ion-selective electrodes [16] may not be reliable in the presence of anesthetics [18].Anesthetics increase the CaM’s Ca^2+^ sensitivity [79,80]. This might have activated many CaM molecules, resulting in significant drop in [Ca^2+^]_i_.Halothane binds to CaM [78].Anesthetics cause Ca^2+^ release via both IP3- and RyR-receptor channels [38].Both caffeine and theophylline increase [Ca^2+^]_i_ by releasing Ca^2+^ from stores [56,57].Heptanol and other uncouplers known to raise [Ca^2+^]_i_ cause gap junction particle crystallization in heart [45,46] and pancreas [47].The interpretation of an extracellular gap junction binding site for anesthetics’ effect on coupling in internally perfused axons and inside-out membrane patches [14,17] is questionable, and may in fact suggest the loss of a cytosolic soluble intermediate (possibly CaM).Lack of cyclic nucleotide role, excluded by experiments with IBMX (a phosphodiesterase inhibitor), forskolin (an activator of adenylate cyclase) or diffusible cAMP and cGMP.Absence of kinase C role, tested with TPA (activator) or H7 (inhibitor).Lack of adenosine-receptor role, tested with adenosine and IBMX treatments.Evidence that the peak of depolarization caused by anesthetics or anesthetics–caffeine matches the peak of Rj rise.Ryanodine-induced inhibition of Rj rise with heptanol–caffeine.There is evidence that the second exposure to heptanol or heptanol–caffeine has a much smaller effect on Rj, suggesting partial depletion of Ca^2+^ stores by the previous exposure.

In view of these findings, while we cannot exclude a direct effect of anesthetics on gap junction proteins and/or lipids, we propose the hypothesis that heptanol, halothane and isoflurane induce gap junction channel gating by releasing Ca^2+^ from internal stores and activating CaM. We realize that this hypothesis needs to be further tested experimentally and it should be stressed that our data were obtained entirely on crayfish axons, which express innexins rather than connexins. Furthermore, there are mysterious data; for example, there is evidence for a strong inhibitory action of 4-aminopyridine on the heptanol effect (Figure 8A). Since 4-aminopyridine is a K^+^ channel blocker, the effect of other K channel blockers should be tested as well. Moreover, the CaM hypothesis needs to be tested with CaM inhibitors, inhibition of CaM expression and overexpression of CaM mutants.

## Figures and Tables

**Figure 1 ijms-23-09017-f001:**
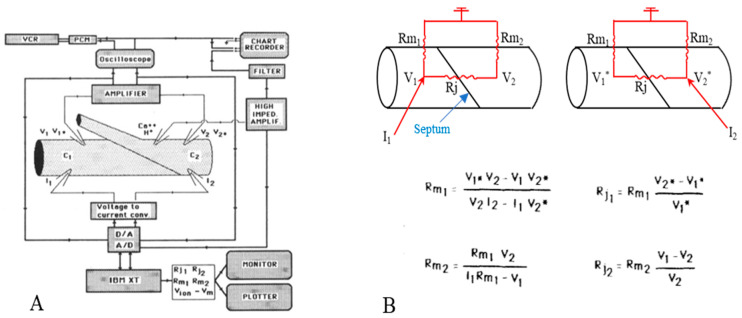
Diagram of electrical recording setup (**A**). Current pulses are injected via current microelectrodes (I_1_, 1_2_) alternatively in the posterior (C_1_) and anterior (C_2_) axon segments. The resulting potentials (V_1_, V_2_, V_1_*, V_2_*) are displayed in the chart recorder and the oscilloscope. The voltage signal (V_ion_), recorded by the ion-selective microelectrode, is displayed in the chart recorder after subtraction of membrane potential (V_m_). (**B**). Equivalent circuit and equations used to calculate junctional (Rj_1_, Rj_2_) and non-junctional (R_m1_, R_m2_) resistances. Adapted from [21].

**Figure 2 ijms-23-09017-f002:**
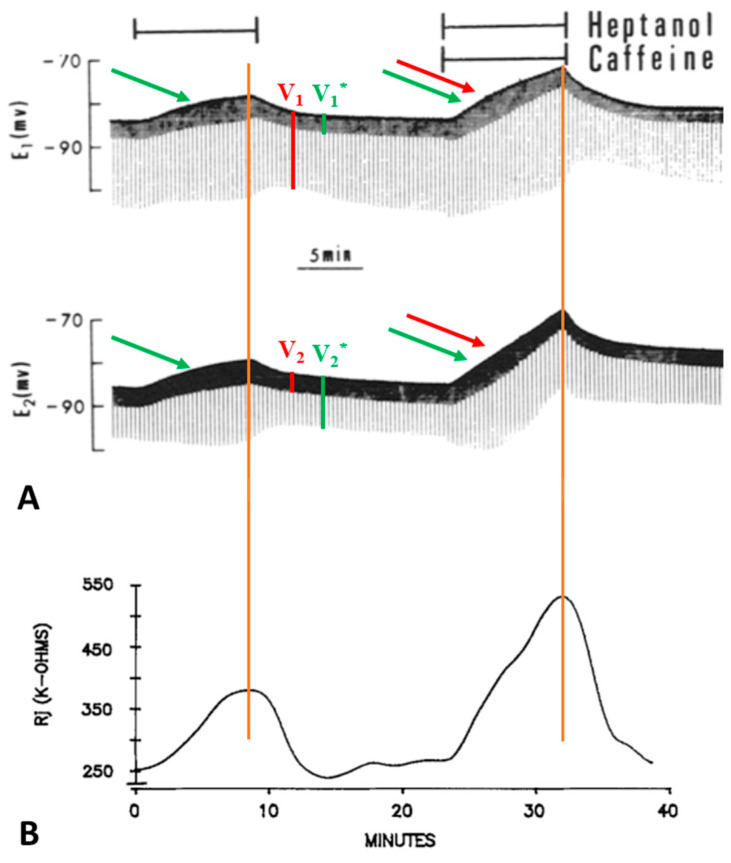
Time course of changes in electrotonic potentials (**A**) and Rj (**B**) in septate axons treated first with heptanol and then with heptanol-caffeine (20 mM caffeine). The Rj recovery rate (**B**) is faster than the onset rate with both heptanol and heptanol-caffeine. Note that the depolarization caused by heptanol ((**A**), green arrow) is greater in the presence of caffeine ((**A**), green and red arrows). Significantly, the peaks of depolarization correspond to the peaks of Rj (vertical red lines). Adapted from [16].

**Figure 3 ijms-23-09017-f003:**
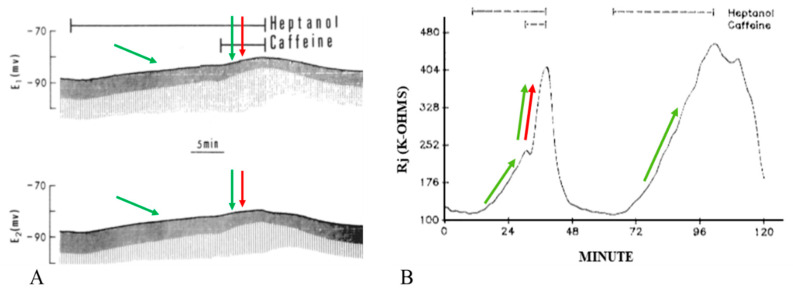
Time course of changes in electrotonic potentials (**A**) and Rj (**B**) in axons uncoupled with heptanol in the presence and absence of 20 mM caffeine. Addition of caffeine to heptanol causes a rapid change in the amplitude of the electrotonic potentials (**A**), indicative of rapid increase in Rj (**B**). Rj increases with heptanol–caffeine 2–3 times as much as with heptanol alone ((**B**), red arrow). The second heptanol treatment shows that the maximal rate of Rj rise with heptanol alone is 14 kΩ/min ((**B**), green arrow), while with heptanol–caffeine (first treatment) the rate more than doubles (35 kΩ/min) ((**B**), red and green arrows). Note that the membrane depolarization caused by heptanol ((**A**), left green arrow) is increased by caffeine addition ((**A**), red and green arrows). The maximum depolarization (**A**) corresponds to the maximum Rj increase (**B**). Adapted from [16].

**Figure 4 ijms-23-09017-f004:**
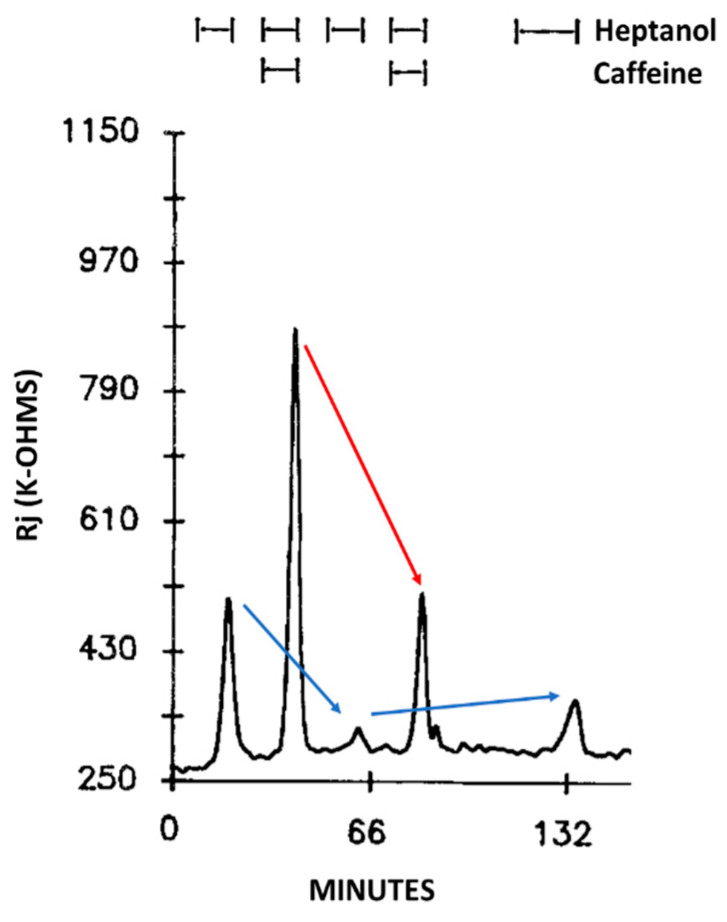
Time course of Rj changes in crayfish axons uncoupled by heptanol in the presence or absence of caffeine. Note the difference in Rj maxima between first and third uncoupling events (left blue arrow), in spite of the same duration of heptanol superfusion (8 min). Similarly, the Rj maxima decrease in the presence of heptanol-caffeine (compare second and forth uncoupling events; red arrow). The heptanol effect on Rj partially recovers (see 5th event, right blue arrow). The drop in Rj maxima in the 2nd and 4th event may indicate the first heptanol and heptanol-caffeine treatments lowered the Ca^2+^ content of the Ca^2+^ stores. Adapted from [16].

**Figure 5 ijms-23-09017-f005:**
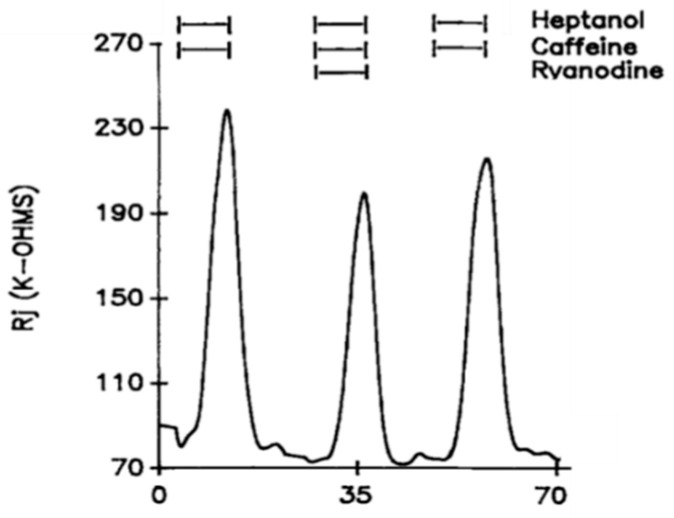
Time course of changes in Rj in crayfish axons uncoupled by heptanol–caffeine (20 mM caffeine) in the presence and absence of ryanodine (10 μM). The caffeine-induced increase in heptanol-uncoupling efficiency is reversibly reduced by the addition of ryanodine. Adapted from [16].

**Figure 6 ijms-23-09017-f006:**
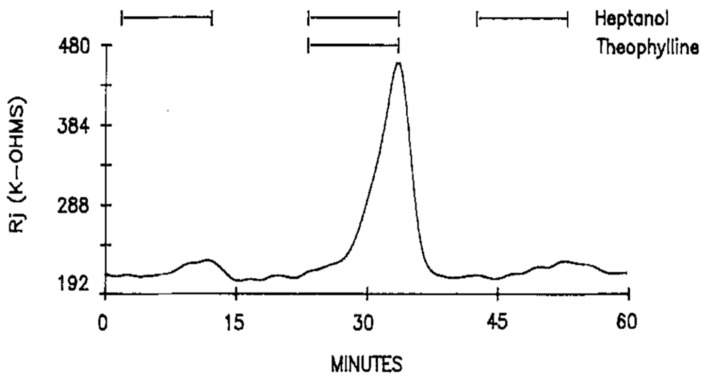
Time course of Rj in crayfish axons uncoupled by heptanol in the presence and absence of 20 mM theophylline. As with caffeine (Figure 2, Figure 3 and Figure 4), theophylline dramatically increases Rj maxima with heptanol. Adapted from [16].

**Figure 7 ijms-23-09017-f007:**
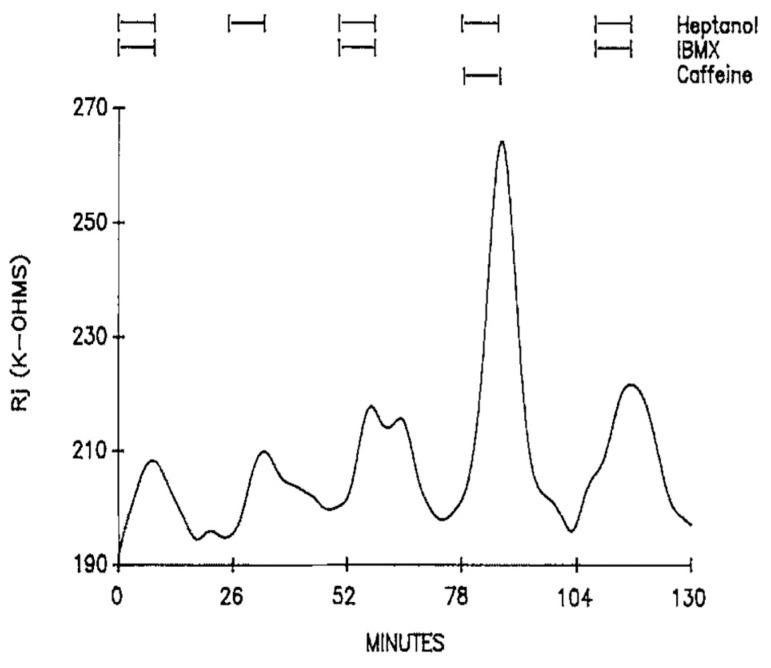
Time course of Rj in crayfish axons uncoupled by heptanol in the presence and absence of 1 mM IBMX. IBMX (an inhibitor of phosphodiesterases and adenosine receptors) does not affect heptanol-induced Rj rise. Adapted from [16].

**Figure 8 ijms-23-09017-f008:**
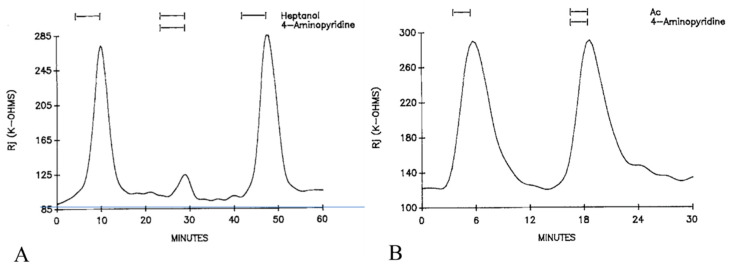
Time course of Rj in crayfish axons uncoupled by heptanol (**A**) or acetate (**B**) in the presence or absence of 5 mM 4-aminopyridine (4-AP). 4-AP dramatically reduce the Rj maximum with heptanol (**A**). In contrast, addition of 4-AP to acetate solutions (Ac) does not affect the acetate-uncoupling efficiency (**B**). Adapted from [16].

**Figure 9 ijms-23-09017-f009:**
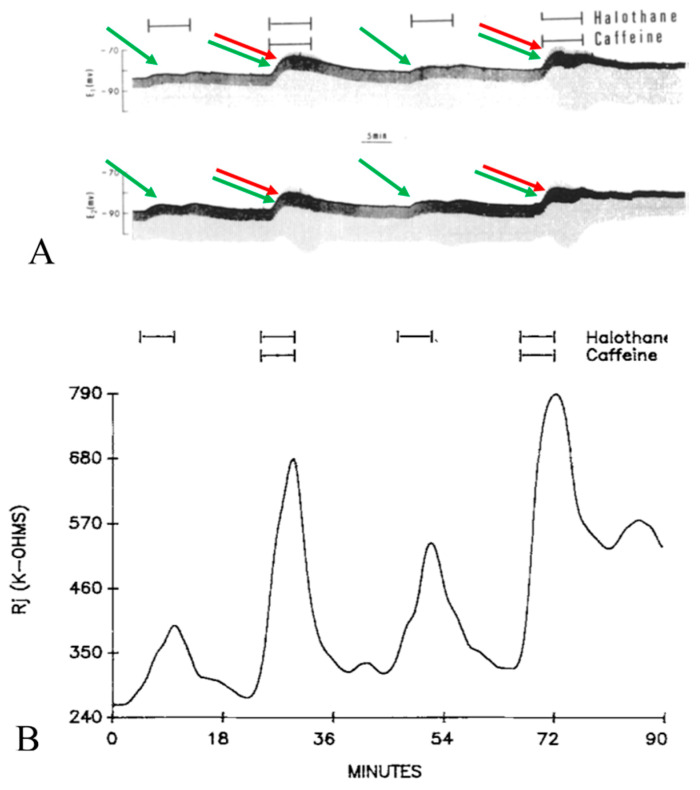
Time course of electrotonic potentials (**A**) and Rj (**B**) in crayfish axons uncoupled by halothane in the presence and absence of 20 mM caffeine. With halothane-caffeine, the increase in V_1_ and V_2_*, and the decrease in V_1_* and V_2_ are greater than with halothane alone ((**A**), red and green arrows), due to a grater increase in Rj (**B**). Note that with caffeine, there is a greater depolarization ((**A**), green and red arrows), whose peak corresponds to the peak or Rj maximum. Following the last halothane-caffeine treatment, coupling recovered only partially (**A**,**B**). Adapted from [16].

**Figure 10 ijms-23-09017-f010:**
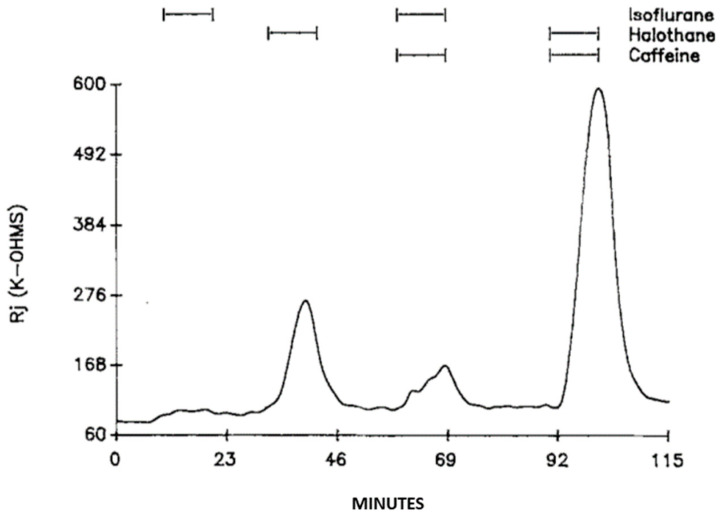
Time course of Rj in crayfish axons treated with either isoflurane or halothane in the presence or absence of 20 mM caffeine. As for heptanol and halothane, caffeine has a pronounced effect on the uncoupling efficiency of isoflurane. However, isoflurane has a significantly weaker effect on Rj than halothane. Adapted from [16].

**Figure 11 ijms-23-09017-f011:**
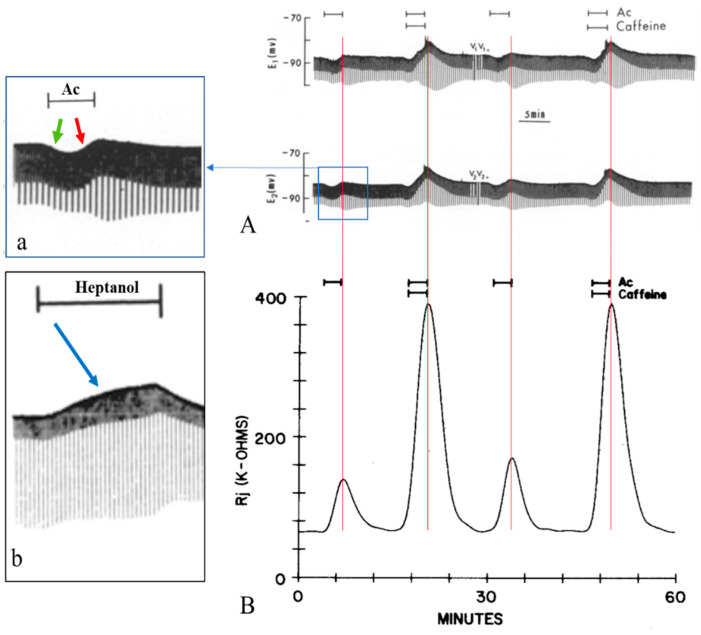
Time course of electrotonic potentials and Rj in crayfish septate axons uncoupled by acetate (Ac) in the presence and absence of 10 mM caffeine (**A**,**B**). (**A**). Low-speed chart recording of membrane and electrotonic potentials. With Ac, V_1_ and V_2_* increase and V_1_* and V_2_ decrease (**A**), reflecting an increase in Rj (**B**). Ac-caffeine causes a larger change in electrotonic potentials (**A**), indicating a larger increase in Rj (**B**). Note that Rj increases with Ac-caffeine 3–4 times as much as with Ac alone (**B**). With Ac or Ac-caffeine, there is a biphasic change in the membrane potential: a moderate hyperpolarization followed by depolarization ((**A**), and inset a, green and red arrows, respectively). In contrast, with heptanol, there is only depolarization (inset b, blue arrow, and Figure 2A and Figure 3A). (**A**,**B**) and inset a were adapted from [20]. Inset b was adapted from [16].

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
