# Peer review of "Anesthetics and Cell–Cell Communication: Potential Ca^2+^-Calmodulin Role in Gap Junction Channel Gating by Heptanol, Halothane and Isoflurane"

_ijms, 2022, doi:10.3390/ijms23169017_

Round 1

Reviewer 1 Report

The manuscript entitled “Anesthetics and Cell-to-cell Communication. Potential Ca2+-Calmodulin Role in GAP JUNCTION Channel Gating by Heptanol, Halothane and Isoflurane” by Camillo Peracchia reevaluates the author’s previous data and conclusions based on experimental data gathered in the upcoming years. The new conclusions are generally in line with the available data and I understand that the author may not be in the possession of experimental environment to perform additional experiments. However, some issues may be discussed further in the manuscript:

11) A compound with unknown mechanism of action can potentially exert its effect on any receptors or channels, present in the cell membrane. Or they may affect the membrane fluidity itself as suggested by other authors. Inhibition of any of these channels may also lead to increased Rj. For example, 4-AP is reported to significantly antagonize the halothane effect, raising the possibility that halothane acts on K+ channels. Why does the author think that the reported anesthetics finally act specifically on connexins?

22) The author investigated the effect of anesthetics in axons. Connexins, however are more abundantly present on astrocytes than on neurons. It would be desirable to discuss whether the conclusion drawn from the data are specific to neuronal connexins or can they be applied generally to all connexin subtypes?

  3) All the experimental interventions inhibited gap junction function. Are there any data in the literature that would strengthen the conclusions from the opposite direction? For example, can intracellular alkalization, that is known to open gap junctions be correlated with CaM function?

Minor issues:

page 10, line 283: EPSP is a much more widely used abbreviation for excitatory postsynaptic potentials than EEPS

There are various typos in the text, which will be hopefully corrected in the final proof.

Author Response

Reviewer #1
The author is grateful to the reviewer for his/her comments that have helped improving this review! The manuscript entitled “Anesthetics and Cell-to-cell Communication. Potential Ca2+-Calmodulin Role in GAP JUNCTION Channel Gating by Heptanol, Halothane and Isoflurane” by Camillo Peracchia reevaluates the author’s previous data and conclusions based on experimental data gathered in the upcoming years. The new conclusions are generally in line with the available data and I understand that the author may not be in the possession of experimental environment to perform additional experiments. However, some issues may be discussed further in the manuscript:
11) A compound with unknown mechanism of action can potentially exert its effect on any receptors or channels, present in the cell membrane. Or they may affect the membrane fluidity itself as suggested by other authors. Agree! See the added sentence starting with “X-ray and neutron diffraction studies of a binary lipid membrane demonstrate that halothane at physiological concentrations produces a pronounced redistribution of lipids between domains of different lipid types identified by different lamellar d-spacings and isotope composition. The redistribution of lipids between domains induced by anesthetics could in principle contribute …..” Inhibition of any of these channels may also lead to increased Rj. For example, 4-AP is reported to significantly antagonize the halothane effect, raising the possibility that halothane acts on K+ channels. The sentence: “Since 4-aminopyridine is a K+-channel blocker, the effect of other K-channel blockers should be tested as well” has been added. Why does the author think that the reported anesthetics finally act specifically on connexins? There is no question that the anesthetics tested cause gap junction channel gating. Since gating is known to result from the action of cytosolic calcium, and the gating effect is potentiated by chemical such as caffeine and theophylline, which increase cytosolic calcium concentration it is reasonable to believe that they ultimately act on gap junction channels, via Ca-CaM and/or affecting the gap junction lipid bilayer. 22) The author investigated the effect of anesthetics in axons. Connexins, however are more abundantly present on astrocytes than on neurons. It would be desirable to discuss whether the conclusion drawn from the data are specific to neuronal connexins or can they be applied generally to all connexin subtypes? True! The following sentence has been added: “and it should be stressed that our data were obtained entirely on crayfish axons which express innexins rather that connexins. Furthermore…”
3) All the experimental interventions inhibited gap junction function. Are there any data in the literature that would strengthen the conclusions from the opposite direction? For example, can intracellular alkalization, that is known to open gap junctions be correlated with CaM function? Whether there is alkalinization caused by anesthetics could not be ascertained since the H+-selective electrodes are unreliable in the presence of anesthetics. Furthermore, cytosolic alkalinization not always increases cell coupling as it has been reported to also increased cytosolic calcium concentration (Li, S. et al., PLoS One, 7, e31905, 2012) and cause uncoupling in various cells (see: Gonzalez-Nieto et al., PNAS 105, 17169-17174, 2008; Loewenstein et al., Journal of General Physiology, 50, 1865-1891, 1967; Rose & Rick, The Journal of Membrane Biology, 44, 377-415. 1978).
Minor issues: page 10, line 283: EPSP is a much more widely used abbreviation for excitatory postsynaptic potentials than EEPS Done There are various typos in the text, which will be hopefully corrected in the final proof. Done

Reviewer 2 Report

The review is well written, but I am not sure if the purpose and aim of this review are explained well in the introduction. I realized and understood them after reading the end of the discussion and summary and conclusion. I recommend improving that in the introduction. I have a few more major issues:

 1. It is not clear based on what the authors selected the studies focusing on anesthetics and cell-to-cell communication and the mechanisms behind that. That must be clear if it is an expert-based selection or a systematic literature search or something between. The authors should clarify that very precisely and explain their strategy very well. Especially if they propose the alternative hypothesis for the cause of the inhibition of GJIC caused by heptanol, halothane, and isoflurane in the brain. The table summarizing all proposed mechanisms for anesthetic-induced GJIC dysregulation would help this review a lot.

2. The most of major findings and images are from one publication [16]. Are there other studies that report the same findings? If so, the authors should stress that. If not, they should discuss that. That confused me a lot.

3. The authors should specify the effect of caffeine and ryanodine on calcium in the ER (L113-125). Similarly, they should also explain why the effect of the co-treatment of heptanol and theophylline on Rj was studied (L150).

4. I guess PKC is missing in this sentence – neither TPA (an activator OF PKC) nor H7 (an inhibitor OF PKC) significantly affects the magnitude of heptanol-induced uncoupling (L167-169).

5. It is not clear extracellular binding site of what is talking about (L313).

6. I would avoid the excessive use of abbreviations. It is not easy to read the text because of that, the flow of the text is disrupted by that.

7. The authors should propose which experiments can resolve this mystery more deeply. That should be the gold outcome of this review. The schema of the proposed mechanism by the authors would also help to illustrate their vision.

Author Response

Reviewer #2
The author is grateful to the reviewer for his/her comments that have helped improving this review! The review is well written, but I am not sure if the purpose and aim of this review are explained well in the introduction. I realized and understood them after reading the end of the discussion and summary and conclusion. I recommend improving that in the introduction. I have a few more major issues: Both summary and introduction have been rewritten to explain the reasons for this review. Basically, we have reconsidered our earlier interpretation of the lack of a calcium role in the gap junction’s gating effect of anesthetics, which relied primarily on the apparent drop in [Ca2+]i. New evidence published in the past 3 decades, evidence for the fact that ion selective electrodes are unreliable in the presence of anesthetics and our earlier data on the potentiation effect of caffeine and theophylline have prompted us to propose the hypothesis of a calcium role in the anesthetics effect on gap junction channel gating. 1. It is not clear based on what the authors selected the studies focusing on anesthetics and cell-to-cell communication and the mechanisms behind that. The reason is that while anesthetics have long been reported to affect direct cell-cell communication, their mechanism of action on gap junctions has never been understood. Most works have considered a direct effect on gap junctions, but no experimental evidence has been produced. That must be clear if it is an expert-based selection or a systematic literature search or something between. The authors should clarify that very precisely and explain their strategy very well. Especially if they propose the alternative hypothesis for the cause of the inhibition of GJIC caused by heptanol, halothane, and isoflurane in the brain. The table summarizing all proposed mechanisms for anesthetic-induced GJIC dysregulation would help this review a lot. We have considered the reviewer’s suggestion of creating a table, but opted not to do it because only two mechanisms have been proposed: one is the role of calcium, which is described in this review and also proposed by Veenstra and DeHaan [33], both based on experimental data, and the other, proposed by several others, but without experimental evidence, which suggests a direct action on the gap junctions’ lipid bilayer. 2. The most of major findings and images are from one publication [16]. Are there other studies that report the same findings? If so, the authors should stress that. If not, they should discuss that. That confused me a lot. Yes, the other study proposing a calcium role is that of Veenstra and DeHaan [33], which is mentioned in detail in this review. 3. The authors should specify the effect of caffeine and ryanodine on calcium in the ER (L113-125). Similarly, they should also explain why the effect of the co-treatment of heptanol and theophylline on Rj was studied (L150). The role of the ER has been specified in more detail in this revised version and 17 new references have been added. Theophylline was used to test the hypothesis that the effect of anesthetics results from phosphodiesterase inhibition. See this modified sentence: The possibility that the effect of caffeine or theophylline is due to an increase in cyclic nucleotides was tested by exposing the axons to 3-isobutyl-1-methylxanthine (IBMX, a phosphodiesterase inhibitor), forskolin (an activator of adenylate cyclase) or diffusible cAMP and cGMP (CPT-cAMP and 8Br-cGMP) [16]. Additions to heptanol of 1 mM IBMX (Figure 7), a phosphodiesterase inhibitor 200 times more potent than caffeine [22], 5 μM forskolin, 500 μM CPT-cAMP, or 200 μM 8Br-cGMP do not significant affect Rj maxima [16]. 4. I guess PKC is missing in this sentence – neither TPA (an activator OF PKC) nor H7 (an inhibitor OF PKC) significantly affects the magnitude of heptanol-induced uncoupling (L167-169).
Done 5. It is not clear extracellular binding site of what is talking about (L313). Changed to: extracellular gap junction binding site. Meaning: extracellular anesthetics’ binding site of gap junction proteins or lipids 6. I would avoid the excessive use of abbreviations. It is not easy to read the text because of that, the flow of the text is disrupted by that. I understand, but it is customary to detail the meaning of acronyms only when they are first mentioned 7. The authors should propose which experiments can resolve this mystery more deeply. That should be the gold outcome of this review. The schema of the proposed mechanism by the authors would also help to illustrate their vision. The last sentence of the review has been modified to include the need to test other K-channel blockers as well.
Since 4-aminopyridine is a K+-channel blocker, the effect of other K-channel blockers should be tested as well. Moreover, the CaM hypothesis needs to be tested with CaM inhibitors, inhibition of CaM expression and overexpression of CaM mutants.

Author Response

Reviewer #3
The author is very grateful to the reviewer for his/her comments that have helped improving significantly this review!
This is a nice hypothesis paper which will steer attention in the gap junction field. The effect of anesthetics on Ca2+-selective electrodes is not well known and this information should be widely disseminated. The paper is well written, very innovative and in order to be improved I would suggest the author to take into consideration the following comments.
Major Comments :
-Role of the IP3 receptor:
l.279 while addition of caffeine or theophylline releases Ca2+ by activating both IP3 and RyR
Bezprozvanny et al. demonstrated caffeine-induced inhibition of inositol(1,4,5)-trisphosphate-(IP3) gated calcium channels from cerebellum incorporated into planar lipid bilayers (PMID: 8186468). Parker and Ivorra found that caffeine inhibits IP3-mediated liberation of intracellular Ca2+ in Xenopus oocytes (PMID: 1844813). Later, Saleem et al. showed that caffeine is a low-affinity antagonist of type 1 IP3 receptors (IP3R1), while it had no significant effect on IP3-evoked Ca2+ release via IP3R2 or IP3R3 (PMID: 24628114).
Joseph et al. (PMID: 24878495) found that isoflurane modulates IP3R channel sensitivity to IP3 only at low subsaturating concentrations of IP3 (<0.1 μM). They also showed that isoflurane causes Ca2+ release from the ER via this activation of IP3R which can regulate intracellular Ca2+ homeostasis and apoptosis.
OK. All done!
-Role of the RyR:
l.264ff . In frog’s skeletal muscle, halothane was found to increase [Ca2+]i by releasing it from the sarcoplasmic reticulum (SR) via the RyR’s Ca2+-release channel
Laver et al. (PMID: 28079567) found that halothane activation of RyR2 is different from that seen in the skeletal isoform RyR1. Unlike RyR1, RyR2 was reported to be responsive to halothane when recorded in bilayers (PMID: 8053596) as well as in myocardial cells (PMID: 1700102, PMID: 3177918).
For gap junction gating in the heart, the cardiac RyR2 isoform is more relevant. It was reported that halothane activated the cardiac ryanodine receptor channel, while isoflurane was ineffective in activating RyR2 (PMID: 28079567, PMID: 8053596).
OK. All done!
-Effect of anesthetics on calmodulin:
l.339 Anesthetics increase the CaM’s Ca2+-sensitivity
Zhou et al. showed that volatile anesthetics (VA) inhibit the activity of calmodulin by interacting with its hydrophobic core (PMID: 22932200). Accordingly, Streiff et al. predicted that volatile anesthetics bind to [Ca(2+)](4)-CaM, but not to apo-CaM. The VA binding sites predicted for the structures of [Ca(2+)](4)-CaM are located in hydrophobic pockets that form when the Ca(2+) binding sites in CaM are saturated (PMID: 16877516).
Volpi et al. reported antagonism of calmodulin by local anesthetics and inhibition of calmodulin-stimulated calcium transport of erythrocyte inside-out membrane vesicle (PMID: 6457977).
Levin and Blanck observed a biphasic effect of halothane and isoflurane on calmodulin, at low concentrations of the anesthetics the affinity of calmodulin for Ca2+ was decreased, while at higher concentrations, the affinity for Ca2+ was increased (PMID: 7604990).
Moreover, Rudnick et al. suggested that halothane mimics calmodulin-blocking agents and may alter CaM interaction with Ca2+-dependent kinases (PMID: 1898840).
OK. All done!
-Effect on halothane on membranes:
l.266 ff. Furthermore, morphological studies also reported changes in gap junction particle size and spacing with heptanol and other uncouplers known to increase [Ca2+]i in heart [45, 46] and pancreas
Halothane was shown to change the domain structure of a model membrane (PMID: 22352350). X-ray and neutron diffraction studies of a binary lipid membrane demonstrate that halothane at physiological concentrations produces a pronounced redistribution of lipids between domains of different lipid types identified by different lamellar d-spacings and isotope composition. The redistribution of lipids between domains induced by anesthetics could in principle contribute to changes in gap junction particle size and spacing.
Plasma membranes of cells show asymmetric lipid distribution between the bilayer leaflets with negative charge of the inner bilayer leaflet (PMID: 4530994). Phospholipid unsaturation is dramatically asymmetric, with the cytoplasmic leaflet being approximately twofold more unsaturated than the exoplasmic leaflet (PMID: 32367017). Atomistic simulations and spectroscopy of leaflet-selective fluorescent probes reveal that the outer PM leaflet is more packed and less diffusive than the inner leaflet. The tightly packed outer leaflet may serve as an effective permeability barrier, while the more fluid inner leaflet may allow for rapid signal transmission. Thus, it is conceivable that the solubility of anesthetics is different in the two halves of the membrane bilayer.
OK. All done!
Minor Comments:
l.32 typo NH2-treminus → NH2-terminus
l.34 Gap junction channels are physiologically regulated
l.51 did not affect
l.58 also to excluded the role of Ca2+i
l.60 past three decades
l.62 trisphosphate
l.67 comma missing after Rj?
l.68+73 superfused
l.120 heptanol-caffeine
l.122 treatment
All corrected
l.154 put comma before nisoldipine
l.164 do not significantly affect
l.213 put comma after However
l.222 Our interpretation was based primarily on
l.223 However, as mentioned in the previous section
l.250 Furthermore, halothane and n-alkanols
l.254 inositol 1,4,5-trisphosphate
l.256 Malignant Hyperthermia
l.259 the halothane induced Ca2+ release(s) from the stores is sensitive to
l.276 Ca2+-induced release(s) from the stores is potentiated by caffeine
l.305 remove . after V1*
l.354 te(r)sted with adenosine
l.365 Figure 7A → should be 8A
Ref 54 spelling error in title: 4-Аminopyridine sequesters intracellular Ca2+ which triggers exocytosis in excitable and non-excitable cells

All corrected

Round 2

Reviewer 3 Report

Please find my comments attached.

Author Response

This review relies heavily on the author’s own work; for a more balanced view alternative interpretations should be presented and evaluated, e.g. regarding the role of cholesterol in gap junctional intercellular communication (GJIC). Cholesterol is a major component of the plasma membrane and enriched in most gap junctions (PMID: 19686581), addition of cholesterol acts on the fluidity of the lipid bilayer and Bastiaanse et al. have shown that heptanol decreases GJIC probably by decreasing the fluidity of cholesterol-rich domains in cardiac cells (PMID: 7508980).

See added paragraph staring with: “Furthermore, gap junctions are rich in. Indeed, Bastiaanse and coworkers have suggested… “

Halothane sensitivity can be influenced by connexin isoform composition. Halothane rapidly and reversibly inhibits GJIC, but does so with junctional channels formed by heteromeric Cx40-Cx43 at significantly lower doses than either of the corresponding homomeric Cx40 or Cx43 channels (PMID: 10850971). How can this be explained in the author’s model?

See added paragraph staring with: “The magnitude of the effect of anesthetics of gap junction channel gating is also related to the type of gap junction protein… “

Further comments:

All corrected

l.19 missing + and ] in ([Ca2i), correct to ([Ca2+]i)

l.59 seemed also to exclude

l.64 replace RyR1 by RyR

l.223 as mentioned in the previous section

l.256 there is still an r missing in Malignant Hypethermia, the correct term is Malignant Hyperthermia

l.269 Ca2+-activated

l.298 typo form  from

l.300 typo depemnds  depends

l.311ff. include ref [44]: In frog’s skeletal muscle, halothane was found to increase [Ca2+]i by releasing it

from the SR via the RyR’s Ca2+-release channel [44]. Later, Laver and coworkers [58] found that halothane

activation of RyR2 is different from that seen in the skeletal isoform RyR1.

l.322 “of” missing: is opposite to that of heptanol

l.323 reference missing: 4-AP fails to induce porcine MH (PMID: 7426246)

l.324 Interestingly, 4-AP has been shown to modify phospholipid metabolism (PMID: 1546958) and to

stimulate protein phosphorylation in a Ca2+-dependent manner (PMID: 6322996).

l.325 there is twice “indirect” in this sentence: Indirect evidence of [Ca2+]i rise with anesthetics is also

indirectly provided by changes in membrane potential (MP)

l.329f. include plural form: Sauviat and coworkers [44] reported that the effects of halothane on membrane

depolarization is are likely to result from

l.373-377 and l.379-383 text is identical

l.398 might be better to write RyR here instead of RyR1, to make it more general (see the part about RyR2

in heart)
